# Getting along to get ahead: The role of social context in tournament promotion and reward systems

George C. Banks[1]*, Christopher E. Whelpley[2], Eean R. Crawford[3], Ernest H. O'Boyle[4], Sven Kepes[2]

**1** Department of Management, University of North Carolina at Charlotte, Charlotte, NC, United States of America, **2** Department of Management and Entrepreneurship, Virginia Commonwealth University, Richmond, VA, United States of America, **3** Department of Management and Entrepreneurship, University of Iowa, Iowa City, IA, United States of America, **4** Department of Management and Entrepreneurship, University of Indiana, Bloomington, IN, United States of America

* gcbanks@gmail.com

**Data Availability Statement:** Data have been posted to the Open Science Framework along with R code. The following study was pre-registered via the Open Science Framework (OSF) at: https://osf.

## Abstract

Tournament theory posits that some organizations are modeled after sports tournaments whereby individuals are incentivized to compete and win against other members of the organization. A persistent criticism of tournament theory is that rank-order success of employees is entirely dependent on *non-interacting* or at least non-cooperating entities. To address what part, if any, cooperation plays in competitive tournaments, this study examines the role of social networks in tournament-style promotion and reward systems. Specifically, we seek to identify the importance of social relationships, such as group dissimilarity, initial tie formation, and tie strength in predicting tournament success. Bringing two largely independent research streams together (one focused on cooperation and one framed around competition), we examine how individuals' performance interacts with their social relationships—their social networks—to influence their chances of winning a tournament. Using the Survivor television series, we analyze the behaviors of 535 interacting contestants across 30 tournaments. In general, the findings help to illustrate how performance and social networks predict tournament advancement. Interestingly, we find that group dissimilarity based on gender, race, and age, largely does not play a role in advancement in the tournaments. Further, the strength of ties fails to mediate between variables such as group dissimilarity and initial tie formation. We conclude by discussing future directions for theoretical and practical exploration of tournament-style promotion systems. Recommendations include continuing to explore and test the role of social dynamics in compensation and promotion systems.

## Introduction

Tournament theory is based on the idea that organizational promotion and reward decisions can be modeled after sports tournaments whereby individuals or teams are incentivized to compete and win against other members of the organization in successive rounds that

io/29rs5/?view_only=
eb710425b9834348b8a30a59db43fd40. The
following study was pre-registered via the Open
Science Framework (OSF) at: https://osf.io/29rs5/?
view_only=
eb710425b9834348b8a30a59db43fd40.

**Funding:** The authors received no specific funding
for this work.

**Competing interests:** The authors have declared
that no competing interests exist.

culminate in an individual or a small number of individuals 'winning' (e.g., receiving promotions, pay, bonuses) [1]. Tournament theory is most applicable in contexts where absolute performance measures are not available or are ineffective for guiding promotion [2, 3]. Specifically, tournaments are more effective when promotion and compensation decisions rely on relative performance evaluations. Though there have been a number of advances in tournament theory since its initial proposal nearly 40 years ago leading to an increase in the popularity of tournaments [2, 4], there remains a number of management-related unknowns and limitations into some facets of tournament-style promotion and reward systems [5].

One particular limitation is that tournament theory has been modeled largely on tournaments where individual performance is synonymous with winning (i.e., performance = output = outcomes), which does not necessarily mirror real life organizations. That is, most work in this area assumes an absolute meritocracy where judges (e.g., supervisors) have full knowledge of each participant's (e.g., subordinate's) performance relative to his or her peers, and interactions are purely competitive [6]. The assumptions of complete knowledge and rationality reflect tournament theory's microeconomic roots, but these assumptions are a substantial divergence from most organizations where social networks, politicking, etc. often play critical roles in reward allocation [7]. Thus, a persistent criticism of the research stream is the attempt to explain rank-order success of employees while treating the contestants as independent, essentially *non-interacting*, entities [5]. In turn, treating interacting tournament participants as independent actors oversimplifies the actual tournament dynamics, likely obfuscating research findings and the reported results [8]. As the common refrain goes, "it isn't what you know, it's who you know." Hence, there is a need to take into consideration social dynamics in tournaments.

This omission is important to address as tournament-style promotion and reward systems in actual organizations are affected by social relations between tournament participants and, therefore, their social networks [5]. The current research seeks to examine the role of social networks in tournament-style promotion and reward systems that, to date, has not been empirically addressed. Specifically, we seek to understand how individuals advance through tournament style promotion and reward systems and the role social networks play in their advancement. Our study incorporates the inherently social, but generally ignored elements of tournament success where tournament success is defined as an individual moving through successive rounds in a tournament. To accomplish this, we identify the importance of social relationships in tournament success and include some relevant characteristics of the social ties, including group dissimilarity and initial tie formation. In addition, we consider the roles of tie strength and individual task performance. Task performance in this context refers to performance behavior rather than whether a participant moves through the tournament. This is consistent with the notion in organizational settings that one's individual task performance is an important contributor to, but not the sole determinant of, whether one receives promotions and rewards. Consequently, we bring two largely independent research streams together to examine how individuals' performance interacts with their social relations to influence their chances of winning the tournament.

Past empirical studies of tournaments have primarily relied on three contexts; professional sports, individual competition game shows, and laboratory experiments. Each has their strengths and weaknesses [5]. To examine the research hypotheses and questions in the current study, we selected the Survivor television show because of the unique characteristics of the show and because it fulfilled five criteria that we found most central to the study of tournaments in the context of social networks. First, the prize must be large enough to mirror that distributed in actual organizations. Second, consistent with other tournament models, the sample should consist of information asymmetries between actors and principals [5]. That is,

both the actors and principals should hold information that the other does not. Third, unlike the sample of past studies on tournaments in which contestants compete as individuals, our sample requires coordination between actors which better mirrors real-life contests in a professional environment. In other words, our sample comes from a context in which social relationships can have a direct and indirect effect on tournament advancement.

Fourth, the sample should be longitudinal to be able to examine potential changes over time and to understand how performance and social relations unfold across tournament rounds. Fifth, variations from contest to contest increase the robustness of the findings if the general framework for a tournament remains largely unchanged. Consequently, the sample should contain multiple iterations of the same or similar tournament to reduce/compensate for the idiosyncrasies of contestants or nuances associated with any individual tournament. For instance, in any single tournament an event could occur that strongly influences the outcomes for individuals competing (e.g., withdrawal of a competitor), but has little to do with criterion relevant tournament factors (e.g., task performance).

## Theoretical framework

In its simplest form, a tournament is a contest between individuals attempting to advance to another hierarchical level within an organization and obtain a corresponding increase in pay, prestige, status, and so on [9]. The idea that organizations can use tournaments as a means of employee promotion and remuneration was first theorized decades ago [2]. This idea was conceptualized in tournament theory, which was developed to propose that promoting and compensating individuals according to their relative ordinal rank in an organization could be as efficient as, if not superior to, more traditional promotion and reward schemes based on absolute individual-level output or performance. The resulting stream of research established the idea of inducing and motivating employee effort to advance in an organization via steeper-than-average pay structures [3, 10]. Accordingly, tournament theory developed, in part, to explain the increasing gap in CEO-to-worker pay [11] and, more generally, the power law distribution of pay across hierarchical levels in organizations [4, 12].

## Components of tournaments

The most fundamental assumption of tournament theory is that individuals are evaluated based on their relative performance rather than their absolute performance. The theory focuses on promotion and reward systems that incentivize individuals to strive for superior performance *relative* to their coworkers instead of more traditional promotion and reward systems that tend to use *absolute* performance as a basis for promotion and reward decisions [5]. As applied to the salary structure for executives, the essential argument is that substantially larger pay associated with promotion to the position of president is not reflective of the executive's higher value to the corporation as president, but rather serves as a reward for being more productive throughout the executive's tenure in more junior positions [4]. Consequently, promotions with large relative pay differences may be associated with small absolute differences in performance.

Although the empirical evidence is still somewhat limited, organizations that use tournament-style promotion and reward systems are believed to accrue two primary benefits when compared with other approaches to promoting and/or rewarding employees (e.g., merit pay, piecemeal). First, tournament-style promotion and reward systems should maximize employee effort by motivating them to compete against their co-workers and achieve the highest possible performance to obtain the desired promotion and associated pay increase. In turn, this

competition should yield higher levels of performance for all participating individuals and the organization as a whole [4, 13].

Second, because individuals are motivated to compete against each other for the highest level of performance, the cost to monitor employee behaviors decreases for the organization [6]. Consequently, managers can reduce their degree of supervision and concentrate on the evaluation of relative employee performance and the resulting rank order. Thus, tournament theory may also alleviate some of the issues associated with the principal-agent problem in organizations [14]. Specifically, tournament-style promotion and reward systems tend to align employee behavior with organizationally prescribed goals by creating a competition among employees in which advancement is based on one's ability to better achieve organizationally relevant goals in comparison with one's peers [13]. This alignment of incentives allows owners (i.e., principals) to reduce their monitoring efforts.

Tournament-style competition is not without its problems [15]. For example, as the rivalry between tournament participants increases, several counterproductive and unethical behaviors could emerge due to the increased emphasis placed on quantifiable relative job performance. Individuals competing in a tournament may not cooperate with one another in the completion of tasks as they seek to gain an edge over their peers, which they tend to view as opponents in a competition rather than colleagues [8, 16]. A lack of cooperation and over-competitiveness can lead to counterproductive acts such as deliberately withholding key information from peers and sabotaging another's work to the detriment of both the individual and organization. Similarly, individuals may alter or manipulate performance information [e.g., through impression management; 17] in an attempt to make themselves appear more favorable relative to their competitors, regardless of the consequences to the organization [18–20]. Taken together, because the winners of tournaments receive promotions and disproportionately large amounts of pay increases or bonuses, incentives exist to motivate individuals to win at all costs despite the potentially adverse effects on their peers and the organization [21].

Third, some research has explored outcomes related to the losers of the tournaments and the potential to decrease the organizational commitment of tournament participants. When tournament theory was originally developed, research focused on tournaments with just one prize (e.g., a promotion). Because the size of the winner's margin is irrelevant in a tournament, just as in a sports tournament (e.g., the difference in prize money between the winner and runner-up in a golf tournament is the same whether the victory is achieved by one shot or by ten), individuals who lose, particularly if it is by a very slim margin, may be inclined to leave the organization [22]. This departure could result in the loss of potentially very high performing individuals [2, 23, 24], which can have adverse effects on organizational functioning and performance.

## The moderating role of social networks in tournaments

A prominent gap in the extant literature pertains to the fact that, in actual organizations, performance relative to all tournament contestants may not be perfectly correlated with tournament success. This is a crucial difference between the organizational setting and the setting where tournament theory has typically been tested. For example, in contexts such as sports tournaments, there are limited, if any, interdependencies between tournament contestants. Hence, task performance is equivalent to tournament success in these contexts. Yet, this perspective fails to take into consideration the role of social networks.

A social network can be defined as the pattern of ties that link individuals within a set of actors [25]. A fundamental tenet of social network theory is that individuals' positions within a network provide opportunities and constraints for their behavior [26]. One way of

conceptualizing an individual's position in the network is in terms of the individual's *tie strength* which ranges from weak to strong and is characterized by the intensity and significance of the relationships between the focal individual and all other organizational actors [27]. Strong ties among individuals involve frequent, close interactions and deep emotional bonds whereas weak ties involve infrequent, constrained interactions and faint or uninvolved emotional bonds [25, 28]. It is worth noting that social ties and political or coalition ties can overlap in a tournament context. While the former pertains to emotional bonds and relationships, the latter involves engagement in behaviors meant to accomplish particular goals using strategies. Both types of ties can factor into one's tie strength.

The opportunities and constraints provided by strong versus weak ties are unique. Strong ties are necessary to facilitate trust, coordination, and the transfer of complex, non-codifiable knowledge, whereas weak ties are useful in the search for a greater range of non-redundant, easily transmitted information [29, 30]. The accumulated strength of one's ties contributes towards one's social capital [i.e., the actual and potential resources derived from one's network; 31] through the amount and quality of information, influence, support and solidarity available to an individual through their social ties.

Social capital may be particularly influential in the relationship between one's task performance and ultimate success in a tournament setting. This is because participants who are competing against each other must also simultaneously work together as coworkers to complete tasks and help the organization to reach its goals [32]. This is different from the types of tournaments that have typically been used in past research (e.g., a golf tournament or a NASCAR race) in which there is no higher-order organizational goal. In other words, to be successful in tournaments in organizational settings, individuals are likely to both work against and cooperate with other tournament members whereby cooperation may manifest in sharing of resources, influence, and solidarity. Conversely, working against one another might lead to a lack of sharing of resources and information, incivility, or even sabotage. Therefore, tournament success includes components of individual task performance relevant to tournament success and social capital from social networks in organizational settings.

We introduce our theoretical model in Fig 1. It is important to note that the appearance of a social network in a tournament is not a random event. Rather, the individuals participating in the tournament tend to proactively manage their relationships and networks in a manner so as to maximize their chances of winning the tournament [33]. In support of this notion, previous research has referred to behaviors in the workplace as being motivated both by the desire to get along and to get ahead [34]. Those who build cooperative partnerships will develop positions of tie strength resulting in available social capital that facilitates advancement in a tournament. Thus, our major theoretical contention is that tournament success (Fig 1, Box 3) will likely be contingent both on one's actual task performance (Fig 1, Box 4) and one's tie strength (Fig 1, Box 2). Our prediction is that, in tournaments where high interdependencies between individual actors exist, individuals who have both high task performance and strong ties will achieve the highest tournament rank.

Past research has illustrated that performance (Fig 1, Box 4) and individual characteristics in isolation from a network are predictive of job promotion [35]. This research focuses on job performance and aligns closely with the *contest-mobility* perspective on promotional opportunity, which suggests that individuals advance based on their own merit. Those with better task performance will consequently be more likely to advance in a tournament setting [32]. Although this line of research seems self-evident (i.e., people who perform better tend to advance further in a tournament), it ignores the role of the social networks in real organizational tournaments [19]. Specifically, individuals with fewer and weaker ties may obtain fewer advancement opportunities and those individuals with more and stronger ties may be

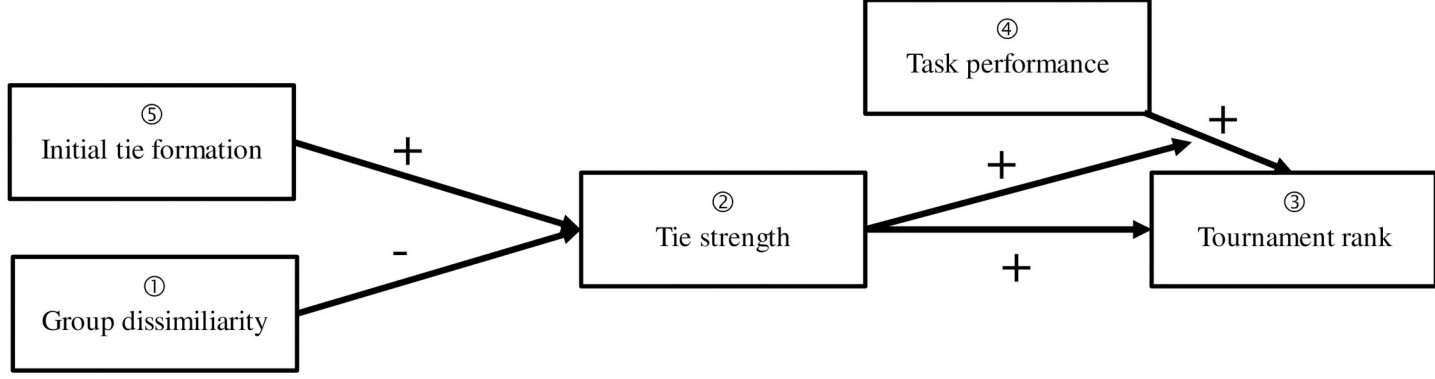

**Fig 1. The role of social networks in tournaments.**

promoted more frequently (see Fig 1, Box 2). The reason for the difference is two-fold. First, there are tangible benefits that accrue to individuals embedded in social networks (e.g., knowledge, information, support, reputation) and, second, managers often do not have an ability to perfectly compare employees [45, 47]. In the organizational context, we know that performance ratings during the appraisal process, the foundation of merit pay systems, are prone to reflect the combined effect of an employee's task performance as well as subjective impressions and considerations of the supervisor [i.e., the rater; 36, 37]. Part of this subjective component is likely dependent on an employee's tie strength, one's accumulated social capital. Furthermore, high levels of social capital allow one to better coordinate with peers, facilitating one's actual job performance. Thus, we expect that individuals with high levels of task performance and strong ties will have the most tournament success.

However, this is not to say that tie strength can substitute for task performance. Actual task performance is still likely to be the predominant component in merit-based pay and merit-based promotions are still the most popular forms of remuneration in organizational settings [38, 39]. For this reason, actual task performance remains critical to advancement in organizations that employ tournaments. Consequently, we expect that individuals with low task performance, but high tie strength will not be able to achieve success to the extent of their high task performance-high tie strength counterparts. Finally, we expect that those with both low performance and low tie strength will advance the least in the tournament for the reasons stated above. Hence, we propose that the strength of one's ties plays a major moderating role in tournament advancement and enhances advancement based on levels of performance. We thus hypothesize the following:

*Hypothesis 1*: The relation between individual's task performance and tournament success will be moderated by tie strength such that the relationship between individual performance and tournament success will be more positive as tie strength increases.

If tie strength plays an important role in affecting the relationship between task performance and tournament success, then it is also important for tournament contestants to understand the sources of strong ties and how they develop. One well-documented influence on tie development is homophily, or the tendency for individuals to affiliate with similar others according to characteristics such as age, gender, and race/ethnicity [40, 41]. Given individuals' predilections for homophily, the social capital created through network status may asymmetrically accrue to those with ties to individuals of similar demographic status. Those who experience greater dissimilarity from the group (see Fig 1, Box 1) may be at a disadvantage in that

social capital (e.g., information, influence, solidarity, and goodwill) will disproportionately accumulate to those who are more similar, demographically, to others in the social network.

In addition to preferences for homophily, information flows more efficiently through demographically similar networks and consequently can be localized within specific groups [53]. This efficiency results in greater access to information and resources for those individuals of a similar demographic background to the group compared to those of different demographic backgrounds [53]. Though demographic differences can be described in different ways, age, gender, and race are typical "fault lines" by which individuals categorize themselves [42]. Consequently, our research focuses on age, gender, and race as the indicators of similarity, and we offer the following:

> *Hypothesis 2*: Tie strength mediates the relation between group dissimilarity and tournament success as ties to those with a more similar demographic profile in terms of age, gender, and race/ethnicity will be more likely to form strong networks and consequently succeed in the tournament.

Another factor affecting the development of ties that is understudied in the tournament literatures, as well as the broader field of management, is the serendipity of initial interactions in new social settings (see Fig 1, Box 5). The social network literature specifically, and sociology literature generally, however, has long documented the influence of propinquity—geographic proximity—on the formation of social relationships. Past scholars [53: 429–430] described, even seemingly trivial factors such as the arrangement of streets [43, 44], dormitory residence halls [45], and legislative seating [46] can influence the formation of relatively weak ties and offer the potential for stronger tie formation. Opportunities to form ties are thus constrained by the social and geographic locations in which individuals are found. Individuals may be co-located in teams or groups (e.g., all applicants hired in the past month will attend the same training; managers assign individuals to work on a specific project; an employee is transferred to a different regional office), in which the initial ties that form have the potential to influence future events even after an individual has made new ties outside of the initial group.

We represent this influence on social relations as the initial ties that individuals form in the early stages of the tournament, constrained by their initial group placement. Both the social processing model [47] and communities of practice [43] literatures provide support for the long-term effects of initial tie formation on future advancement. Specifically, the social processing model suggests that when individuals evaluate a situation, social mechanisms influence their behaviors, opinions, and knowledge of others with whom they associate [47]. Consequently, individuals are more likely to act and work similarly to those with whom they formed initial ties, which increases their tie strength. In turn, tie strength affects an individual's overall tournament advancement and success.

A similar prediction regarding the long term effects of initial tie formation can be found within the communities of practice literature [48, 49]. Here, the types of knowledge shared through initial work ties are relevant to ultimate tournament success. Communities (i.e., networks; see Fig 2) share information with one another and depending on the learned knowledge of the community and community members' willingness to share information, tournament success can be improved [50]. Taken together, both the social processing model and work related to communities of practice, suggest that initial tie formation will influence tie strength and ultimately, one's tournament success.

> *Hypothesis 3*: The relation between initial tie formation and tournament success will be mediated by tie strength such that initial relationship formation will lead to stronger ties which ultimately leads to greater tournament success.

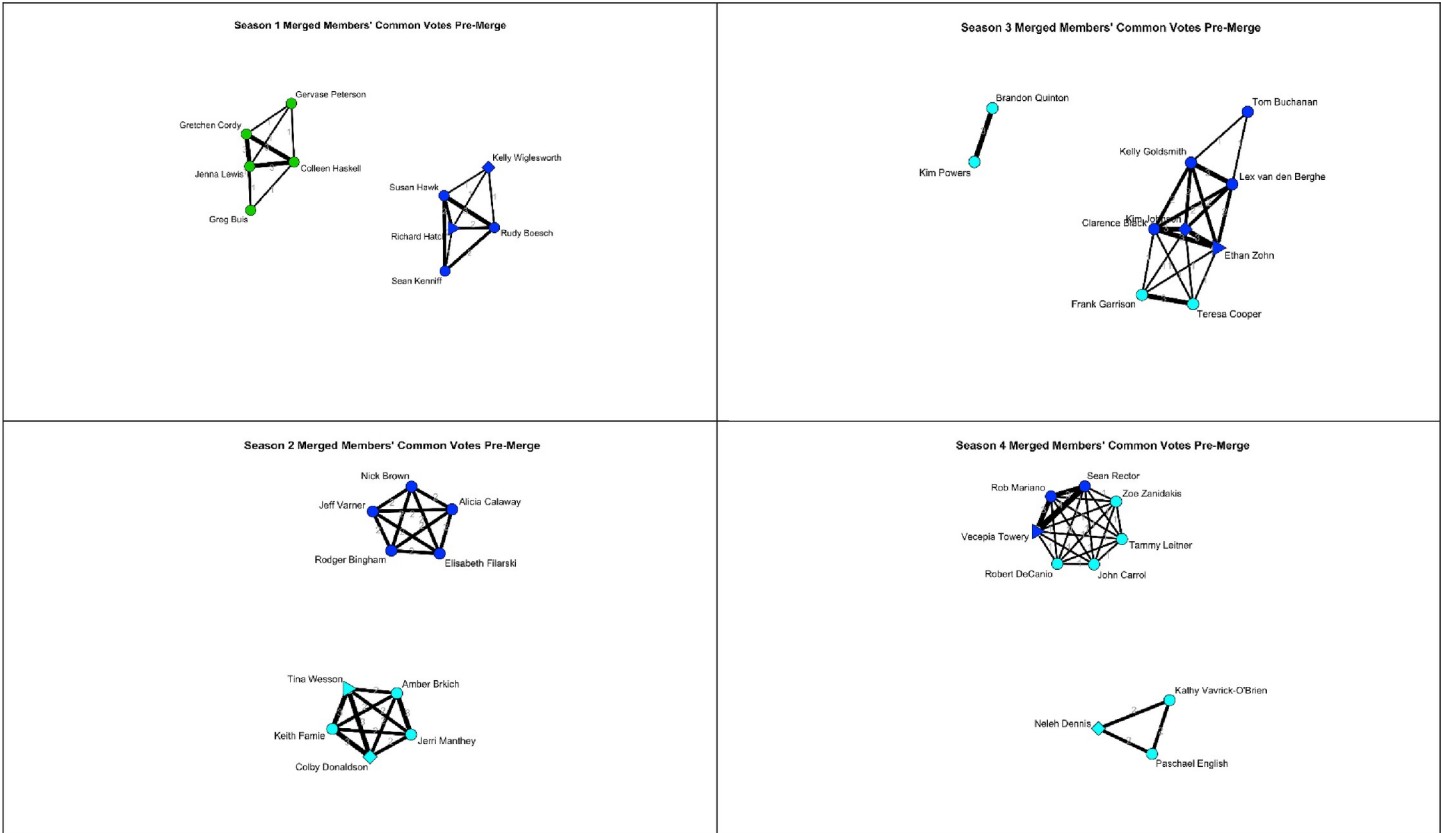

*Note*: Vertices are colored by original team membership; all vertex shapes are circles, except square for runner-up contestant, and triangle for winner; edge values are the number of common votes cast prior to the merge; edge weight (thickness) corresponds to number of common votes cast

**Fig 2. The role of social networks in tournaments.**

A research question that is yet to be examined is how the demographic profiles of those with whom one associates change over time in a tournament setting. On the one hand, individuals may have a preference to associate themselves with and, thus, form strong ties with individuals of a similar demographic profile [53]. Consequently, it is possible that networks in a tournament setting will become more homogenous over time. On the other hand, individuals attempting to win a tournament may prefer to associate themselves with individuals who maximize their probability for success, regardless of their demographic similarity. Consequently, the literature is unclear about how network demographics in group settings evolve across time. In other words, do group characteristics become more similar to winners? We thus ask:

*Research question 1*: Do groups become more homogeneous over time?

Another research question that has yet to be addressed in the literature is exactly how tournament winners progress through a tournament with multiple rounds in an organizational context where social relations play an important role in who wins individual rounds in the tournament. For instance, winners of early tournament rounds may form early ties with other participants and maintain and develop them further throughout the following rounds of the tournament. These early formed ties could be used to form alliances and increase the likelihood of tournament success in subsequent rounds. Only after rivals have been eliminated

might members in a group focus on outperforming each other. Or it is possible that formed alliances dissolve quickly as a result of internal competition and after which "free-agents" attempt to leverage the opportunity to cooperate with members of outside groups. It is interesting to know *how* tournament contestants choose to cooperate and in particular, how they respond to those whom they have formed ties with compared to those in outside groups. Hence, we wish to understand exactly how tournament winners navigate a social network in a competitive tournament context. Can first and second place finishers be predicted by formed alliances? We therefore ask:

> *Research question 2*: Do formed alliances tend to eliminate other cohesive out-groups first and then proceed to turn on each other?

Another way to study the progression of tournament winners in a tournament where there are social elements is to understand the role of early group cohesion (i.e., the extent to which groups are united or stick together). As mentioned above, tournament winners may progress through a tournament by drawing upon the strength of their immediate network. We thus propose that when the cohesion of an individual's group early in a tournament is weak, this lack of cohesion may hurt the individual's chances of tournament success. Therefore, we are curious to explore if tournament contestants embedded in more cohesive groups are more likely to experience greater tournament success than less cohesive groups. More formally, we ask:

> *Research question 3*: Does early group cohesion play a role in the success of tournament winners?

## Methods

The following study was pre-registered via the Open Science Framework (OSF) at: https://osf. io/29rs5/?view_only=eb710425b9834348b8a30a59db43fd40. Due to theoretical considerations and conversations between the authors, the order of the hypotheses was changed and there were some slight edits to the wording of the hypotheses. For example, based on discussions, we reconceptualized tie strength as the moderator of the performance-tournament rank relation rather than performance as the moderator. The full dataset and R script for analysis are also available on the OSF (https://osf.io/yws8x/?view_only=9246647955cb45de86047a9d1e86e85c).

## Sample

Our sample was derived from participants on the Columbia Broadcasting System (CBS) reality TV show Survivor, which, as noted, gives us a unique view into the social aspects of a tournament setting that we feel is superior to other tournament research that uses sports data [e.g., 51, 52] or game show contexts that lack a social interaction component, such as *Jeopardy* [e.g., 53].

Survivor is a contest-based tournament in which 16–20 adults live in a wilderness area, usually a tropical island location, to survive for 39 days with only the barest of provisions. Initially the contestants are grouped into 2–4 teams that live in separate locations. About once per day or every other day, the teams are brought to a common area to compete head-to-head in physical (e.g., stamina events including swimming, running, lifting, and balancing) and mental (e.g., puzzles, memory tests) contests. In one type of contest, called a "reward challenge," the winning team receives rewards such as additional food rations, comforts to improve life at camp, or adventure excursions, whereas the losing team receives nothing. In another type of contest, called an "immunity challenge," the winning team is protected from having to vote a team member out of the tournament and the losing team must enter a council that evening to

vote a team member out of the tournament. The number of team members competing in each challenge is equalized (i.e., the team(s) with more surviving contestants are only allowed to have as many members compete in the challenge as the smallest remaining team). The tournament proceeds over the course of 14–21 rounds, each concluded by the council in which a team member is voted out.

A significant midpoint in the tournament rounds is known as the "merge"—the time at which the initial 2–4 teams, having been winnowed by voting out their members in prior councils, are merged. At this point, the team competitions shift to individual competitions where the remaining 10–12 contestants compete in individual challenges. Individual reward challenge winners similarly receive food, comfort, or adventure rewards. Individual immunity challenge winners receive protection from being voted out at the next council. All team members participate in every post-merge council until they themselves are voted out. The contest is won in a final council in which jury members (former contestants) vote for the winner among the last 2–3 remaining contestants.

Data were obtained from archival internet sources by recording the descriptive statistics of each contestant, their performance in the tournament, as well as the voting matrices for each tournament (www.cbs.com/survivor; www.wikipedia.org/wiki/Survivor; https://survivor.fandom.com/wiki/). At the time of pre-registration, 30 tournaments had been conducted which allowed for a total sample size of n = 535. Results of ICC1 calculations indicated that scores on key variables (e.g., performance, tie strength) could not be predicted by tournament (ICC1< .10). In total, there were 50% males and 50% females. The median age was 33 (minimum = 19; max = 75). Caucasians made up 81% of the sample.

## Measures

**Task performance.** We operationalized individual task performance as individual contestants' number of wins in the physical (e.g., a stamina event) and mental (e.g., a puzzle or memorization) contests. As noted above, contests occurring prior to the merge were largely team-based, whereas contests following the merge were largely individual contests. Thus, we were faced with a decision as to whether to include team- and individual-based contests in the performance metric. We chose only individual contests as the focus of the study is on the performance of the individual tournament contestants. This choice reduces our final sample to the number of contestants who remained in the tournament until the merge ($N = 313$). There were two sub-types of performance: reward and immunity. The combined individual performance score (reward and immunity), individual reward-only, and individual immunity-only were correlated with tournament rank at .51, .46, and .38, respectively.

**Tie strength.** There are many potential ways to index the strength of a social relationship between contestants in a Survivor tournament. An objective manifestation of a relationship existing between contestants is for both to cast a vote for the same person at a council (see Fig 2). Therefore, we operationalized a tie between individuals as having cast a vote in common at a council, and we operationalized tie strength as the sum of a contestant's common votes cast with all other members in the tournament. Tie strength was normalized by the number of votes. Thus, the measure of tie strength can be interpreted as the number of other people that voted the same way as the focal contestant given each opportunity to vote. Tie strength between contestants increased even if their common vote was in the minority (i.e., the contestant they voted for was not eliminated in that round). We note that we exclude the final vote for the winner in the tie strength calculation. The final vote is made by the eliminated contestants comprising the jury rather than the remaining finalists of the tournament. Further, this final vote creates predictor-criterion overlap because it directly determines the winner and

runner ups. Additional sensitivity analyses are available upon request or by utilizing the open data provided on the OSF.

**Group dissimilarity.**   We calculated group dissimilarity based on demographic characteristics (e.g., age, gender, race) [54, 55] using Euclidean distance with the following formula to create an aggregated distance score across all three demographic traits:

$$d_{i,j} = \sqrt{\sum_{k=1}^{p} (x_{ik} - x_{jk})^2}$$

**Initial tie formation.**   We calculated initial tie formation as the sum of common votes a person cast prior to the merge with only members in their initial team. This sum is also affected by the number of opportunities to cast a vote; thus, we normalized it for each contestant by dividing it by the total number of votes the person cast before the merge. Therefore, given each opportunity to vote prior to the merge, a low number suggests that a contestant cast few votes in common with members of their initial team before the merge, reflective of the formation of few initial ties. A high number suggests that a contestant cast many votes in common with other members of their initial team before the merge, and thus formed several initial ties.

**Formed alliances.**   In addition to the variables measured in our model, we also ask research questions about the relations of other important tournament variables. As mentioned previously, a critical point in the tournament is when the initial teams are merged into one. This point is interesting because the merge does not instantly erase the boundaries between two or more formerly distinct teams. Individuals consider to what extent they wish to maintain prior relationships with initial team members while simultaneously considering opportunities to form new relationships with former adversaries. Members of initial teams may have incentives to stay banded together to have more voting power as an alliance. However, disaffected members of an initial team who have lost considerable status (for example, by having been on the losing side of votes leading up to the merge) also have potential to gain voting power by floating their allegiance to form new alliances. The decision of how individuals choose to align themselves is most clearly manifested in the votes cast at the first council following the merge. This council is usually the largest of the entire tournament and provides the most opportunities for contestants to align themselves with a voting bloc of sufficient size and power to eliminate other rivals.

Accordingly, we operationalized formed alliances as the size of the voting bloc each member belonged to based on the first post-merge vote cast. A voting bloc was determined by the number of people voting for a common individual in the vote. The size of voting blocs in our data ranges from 1—occurring when a contestant made a vote for someone that no one else voted for and thus had no vote in common with anyone—to 10—occurring when twelve members made the merge, and all but two voted the same person out.

**Early group cohesion.**   We also based our measure of early group cohesion on votes cast, but this time accounting for the similarity in voting patterns within the initial teams. Greater cohesion manifests in initial teams whose members all vote similarly, as opposed to initial teams in which individuals or subgroups vote for different individuals. Simply put, early group cohesion represents the extent to which early votes within competing teams were unanimous. Early group cohesion may increase the likelihood that members remain together as a cohesive alliance in subsequent rounds of the tournament. However, early group cohesion is not synonymous with the creation of a future alliance in subsequent rounds as individuals may find it advantageous to depart from a cohesive but smaller group to join with a larger voting bloc. To examine this, we constrained our measure of early group cohesion as the closeness and unity

among original team members who were still present in the tournament at the point of the merge. For these individuals, we calculated early group cohesion as the number of common votes that merged members cast with members of their original team prior to the merge, divided by the maximum possible number of votes they could have cast in common.

This calculation is a simple adaptation of the traditional density calculation in social network research where cohesion is indexed as the number of observed ties (denoted as $t$) divided by the maximum possible number of ties (calculated as $n * (n—1) / 2$, where $n$ is the number of actors in the network). In our case, the density numerator is the number of common votes cast (denoted as $t$), and the density denominator is the maximum possible number of common votes that could have been cast (calculated as $n * (n—1) / 2 * v$, where $n$ is the number of original team members that made the merge, and $v$ is the maximum number of pre-merge votes the original team members cast. This measure of early group cohesion ranges from 0.0—occurring when individuals who made the merge never cast a common vote with any of their other team members who also made the merge, to 1.0—occurring when all members of an original team making the merge always voted the same. There are four instances where early group cohesion could not be calculated and thus is treated as a missing value. The first three occurred when an individual was the only one from their original team to make the merge, and thus had no one from their original team with whom they could have cast a vote in common remaining in the tournament. In these cases, the denominator is zero because $n = 1$ (and thus $1 * [1–1]$ is equal to zero). The other instance occurred when an entire team made it to the merge without ever losing and having to vote someone out. In this case the numerator and denominator are zero because they had no votes in common ($t = 0$) because they never had to vote ($v = 0$).

**Outcome and control variables.**   The primary outcome variable was tournament success in the form of one's final rank. Tournament rank was reversed scored so that higher scores indicate that an individual advanced further in the tournament. Following best practice guidance on control variables [56, 57], tournament size was also considered as a control variable based on suggestions from past research that tournament size affects the dynamics of tournament participants, such as effort level exerted [5]. Tournament size can also affect both one's probability of advancing further in the tournament as well as the number of contestants with which one can form ties. Composition of the final jury was originally proposed as a control variable; however, this control variable was no longer was necessary after the final vote was excluded from the calculation of tie strength to avoid concerns about predictor-criterion overlap.

## Results

Table 1 displays the descriptive statistics and correlations for all the variables in this study. There are several correlations of note. For example, tie strength was correlated with tournament rank ($r = 0.12$; $p = .029$), which suggests that as one develops stronger relationships with others, one also tends to advance farther in the tournament. In addition, performance was related to tournament rank, suggesting that as one's performance increased, the final ranking improved ($r = 0.17$; $p = .003$). Initial tie formation was related to tournament rank ($r = 0.42$; $p < .001$), but, surprisingly, group dissimilarity was not ($r = -0.06$; $p = .150$). Further, when considering individual demographic variables, age ($r = -0.05$; $p = .290$), gender ($r = 0.07$; $p = .113$), and racial minority status ($r = -0.02$; $p = .601$) did not relate to tournament success. Thus, by looking at the bivariate correlations some interesting patterns start to emerge.

### Hypothesis testing

Hypothesis 1 stated that the relation between task performance and tournament rank will be moderated by an individual's tie strength such that higher levels of tie strength will strengthen

**Table 1. Means, standard deviations, and correlations with confidence intervals.**

| Variable | Mean | SD | 1 | 2 | 3 | 4 | 5 | 6 | 7 | 8 |
|---|---|---|---|---|---|---|---|---|---|---|---|
| 1. Age | 33.28 | 10.31 | | | | | | | | |
| 2. Racial minority | - | - | -.03 | | | | | | | |
| | | | $p = .421$ | | | | | | | |
| | | | [-.12, .05] | | | | | | | |
| 3. Gender | - | - | .12 | -.02 | | | | | | |
| | | | $p = .007$ | $p = .616$ | | | | | | |
| | | | [.03, .20] | [-.11, .06] | | | | | | |
| 4. Group dissimilarity | 0.41 | 0.21 | .29 | .45 | .01 | | | | | |
| | | | $p < .001$ | $p < .001$ | $p = .738$ | | | | | |
| | | | [.21, .37] | [.38, .51] | [-.07, .10] | | | | | |
| 5. Tie Strength | 3.18 | 1.13 | -.01 | .02 | -.03 | .03 | | | | |
| | | | $p = .906$ | $p = .727$ | $p = .545$ | $p = .658$ | | | | |
| | | | [-.12, .10] | [-.09, .13] | [-.15, .08] | [-.09, .14] | | | | |
| 6. Initial Tie Formation | 3.23 | 1.62 | -.05 | -.03 | .08 | -.03 | .06 | | | |
| | | | $p = .262$ | $p = .428$ | $p = .058$ | $p = .559$ | $p = .314$ | | | |
| | | | [-.13, .04] | [-.12, .05] | [-.00, .17] | [-.11, .06] | [-.05, .17] | | | |
| 7. Performance | 0.18 | 0.17 | -.04 | -.04 | .14 | -.05 | .06 | -.04 | | |
| | | | $p = .476$ | $p = .476$ | $p = .011$ | $p = .380$ | $p = .315$ | $p = .510$ | | |
| | | | [-.15, .07] | [-.15, .07] | [.03, .25] | [-.16, .06] | [-.05, .17] | [-.15, .07] | | |
| 8. Tournament rank | 11.52 | 5.25 | -.05 | -.02 | .07 | -.06 | .12 | .42 | .17 | |
| | | | $p = .290$ | $p = .601$ | $p = .113$ | $p = .150$ | $p = .029$ | $p < .001$ | $p = .003$ | |
| | | | [-.13, .04] | [-.11, .06] | [-.02, .15] | [-.15, .02] | [.01, .23] | [.35, .49] | [.06, .27] | |
| 9. Tournament size | 18.03 | 1.55 | -.00 | .10 | -.01 | .11 | .15 | .21 | .07 | -.14 |
| | | | $p = .978$ | $p = .023$ | $p = .758$ | $p = .014$ | $p = .007$ | $p < .001$ | $p = .239$ | $p = .001$ |
| | | | [-.09, .08] | [.01, .18] | [-.10, .07] | [.02, .19] | [.04, .26] | [.12, .29] | [-.04, .18] | [-.22, -.06] |

[a] *SD*: Standard deviation $N = 313\sim535$. For sex, male = 1, female = 0; For racial minority, racial minority = 1; non-racial minority = 0. Exact *p*-values provided (two-tailed tests). Estimates square brackets are the 95% confidence interval for each correlation.

the relation between performance and tournament success. An interaction term was calculated based on the mean-centered variables of task performance and tie strength. The results of this test are displayed in Table 2. Model 1a is a baseline model that controls for tournament size. In Model 2a, task performance and tie strength were entered leading to $\Delta R^2 = .043$. In Model 3a, the interaction term was entered illustrating a small and statistically significant change in $R^2$ ($\Delta R^2 = .020$). We graph the nature of the interaction effect in Fig 3. Tournament advancement was at its highest when tie strength was low and performance was high. Thus, Hypothesis 1 was not supported as the nature of the interaction effect was not as predicted.

Next, we tested Hypothesis 2, which stated that tie strength mediates the relation between group dissimilarity and tournament success as ties to those with a more similar demographic profile in terms of age, gender, and race/ethnicity will be more likely to form strong networks and consequently succeed in the tournament. Again using hierarchical regression, we tested a mediation analysis for hypothesis 3, which stated that the relation between initial tie formation and tournament success will be mediated by tie strength such that initial relationship formation will lead to stronger ties which ultimately leads to greater tournament success.

The relevant findings are presented in Table 2. Tournament size was again used as a control variable in all analyses. In Model 2b, tournament rank is regressed onto group dissimilarity ($\beta = -.05$, $p = .266$). In Model 2c, tie strength is regressed onto group dissimilarity ($\beta = .00$, $p = .942$).

**Table 2. Regression analyses.**

| Hypothesis 1 | | | | | | | | | |
|---|---|---|---|---|---|---|---|---|---|
| **Tournament rank** | **Model 1a** | | | **Model 2a** | | | **Model 3a** | | |
| | **B** | **SE** | **β** | **B** | **SE** | **β** | **B** | **SE** | **β** |
| Tournament size | -.47 ($p$ = .001; $t$ = -3.26) | .15 | -.14 | -.19 ($p$ = .100; $t$ = -1.65) | .11 | -.09 | -.16 ($p$ = .169; $t$ = -1.38) | .11 | -.08 |
| Performance | | | | 2.86 ($p$ = .005; $t$ = 2.83) | 1.01 | .16 | 2.31 ($p$ = .025; $t$ = 2.26) | 1.03 | .13 |
| Tie strength | | | | .35 ($p$ = .023; $t$ = 2.29) | .15 | .13 | .50 ($p$ = .002; $t$ = 3.08) | .15 | .19 |
| Performance x Tie strength | | | | | | | -2.09 ($p$ = .012; $t$ = -2.52) | .83 | -.15 |
| **ΔR²** | | | | | | .043 | | | .020 |
| **R²** | | | .019 | | | .047 | | | .067 |

| Hypothesis 2 | | | | | | | | | |
|---|---|---|---|---|---|---|---|---|---|
| **Tournament rank** | **Model 1b** | | | **Model 2b** | | | **Model 3b** | | |
| | **B** | **SE** | **β** | **B** | **SE** | **β** | **B** | **SE** | **β** |
| Tournament size | -.47 ($p$ = .001; $t$ = -3.26) | .15 | -.14 | -.46 ($p$ = .001; $t$ = -3.12) | .15 | -.13 | -.18 ($p$ = .121; $t$ = -1.56) | .12 | -.09 |
| Group dissimilarity | | | | -1.21 ($p$ = .266; $t$ = -1.11) | 1.09 | -.05 | .59 ($p$ = .497; $t$ = .68) | .87 | .04 |
| Tie strength | | | | | | | .37 ($p$ = .017; $t$ = 2.40) | .16 | .14 |
| **ΔR²** | | | | | | .003 | | | .002 |
| **R²** | | | .019 | | | .022 | | | .024 |
| **Tie Strength** | **Model 1c** | | | **Model 2c** | | | | | |
| | **B** | **SE** | **β** | **B** | **SE** | **β** | | | |
| Tournament size | .11 ($p$ = .007; $t$ = 2.73) | .04 | .15 | .11 ($p$ = .008; $t$ = 2.69) | .04 | .15 | | | |
| Group dissimilarity | | | | .02 ($p$ = .942; $t$ = 0.07) | .32 | .00 | | | |
| **ΔR²** | | | | | | .001 | | | |
| **R²** | | | .023 | | | .024 | | | |

| Hypothesis 3 | | | | | | | | | |
|---|---|---|---|---|---|---|---|---|---|
| **Tournament rank** | **Model 1d** | | | **Model 2d** | | | **Model 3d** | | |
| | **B** | **SE** | **β** | **B** | **SE** | **β** | **B** | **SE** | **β** |
| Tournament size | -.47 ($p$ = .001; $t$ = -3.26) | .15 | -.14 | -.79 ($p$ = .000; $t$ = 9.00) | .13 | -.24 | -.12 ($p$ = .330; $t$ = -.98) | .11 | -.06 |
| Initial tie formation | | | | 1.54 ($p$ = .000; $t$ = 12.07) | .13 | .47 | -.20 ($p$ = .133; $t$ = -1.50) | .13 | -.09 |
| Tie strength | | | | | | | .35 ($p$ = .025; $t$ = 2.25) | .16 | .13 |
| **ΔR²** | | | | | | .215 | | | .000 |
| **R²** | | | .019 | | | .027 | | | .027 |
| **Tie strength** | **Model 1e** | | | **Model 2e** | | | | | |
| | **B** | **SE** | **β** | **B** | **SE** | **β** | | | |
| Tournament size | .11 ($p$ = .007; $t$ = 2.73) | .04 | .15 | .11 ($p$ = .012 $t$ = 2.53) | .04 | .15 | | | |
| Initial tie formation | | | | .01 ($p$ = .802; $t$ = .250) | .04 | .01 | | | |
| **ΔR²** | | | | | | .001 | | | |
| **R²** | | | .023 | | | .024 | | | |

*Note.* B = unstandardized regression coefficient, SE = standard error, β = standardized regression coefficient. All predictors are mean-centered; $\Delta R^2$ = Change in $R^2$ between models; Exact $p$-values provided (two-tailed tests).

Finally, in Model 3b, tournament rank is regressed on both group dissimilarity ($\beta$ = .04, $p$ = .497) and tie strength ($\beta$ = .14, $p$ = .017). In sum, there does not appear to be evidence that group dissimilarity predicts the mediator tie strength and thus, no mediation effect was detected. Though null, we find this result compelling and will explore it further in the discussion section. To test Hypothesis 3, in Model 2d tournament rank is regressed onto initial tie formation ($\beta$ = .47, $p$ = .000). Next, in Model 2e, tie strength is regressed onto initial tie formation

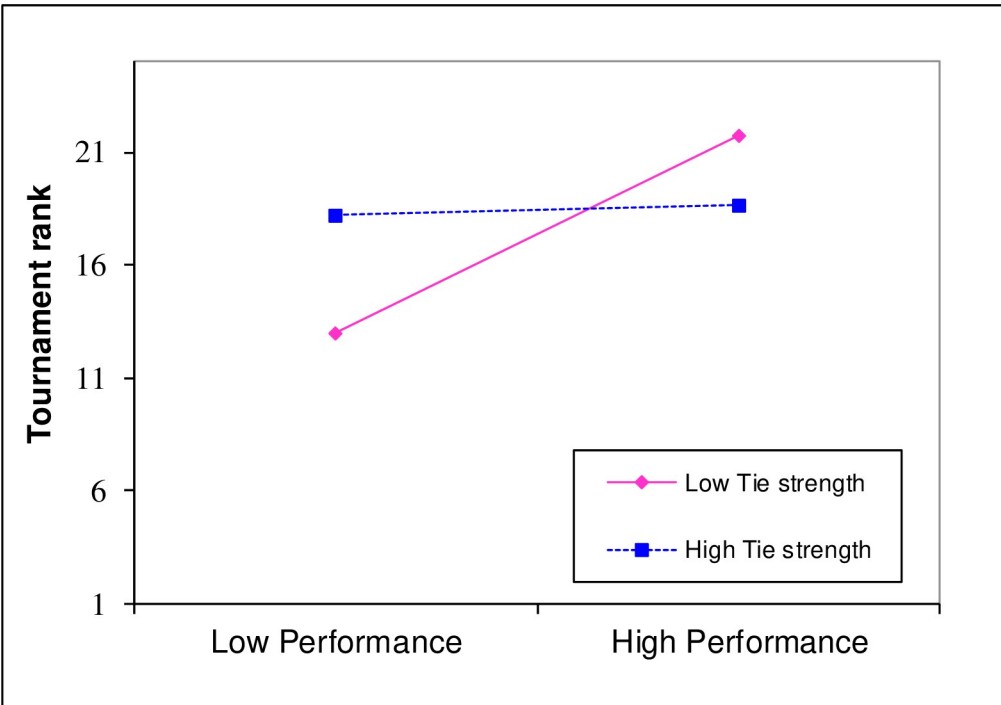

**Fig 3. The interaction of performance and tie strength.**

($\beta$ = .01, $p$ = .802). Finally, in Model 3d, tournament rank is regressed simultaneously onto initial tie formation ($\beta$ = -.09, $p$ = .133) and tie strength ($\beta$ = .13, $p$ = .025). Consequently, again a mediation effect was not found as initial tie formation did not predict tie strength.

### Research questions

**Research question 1.** Next, we consider our research questions. We first asked whether groups become more homogeneous over time? In other words, do group characteristics become more similar to the first and second place winners? To accomplish this, we considered the demographic characteristics of age and sex. Given that only 19% of the sample included racial minorities it was determined that there were not sufficient data to consider the role of this characteristic. To evaluate how the age of the groups changed throughout the course of the tournaments, we first calculated the average age of the tournament contestants at time one, not including the tournament winner, and then calculated the absolute difference between the average age and the age of the tournament winner. We then repeated this process throughout each tournament round to observe the pattern.

The mean and median age of tournament winners was 32 years old (minimum = 21; maximum = 57). In general, there did seem to be a lot of variance. For instance, many tournaments had an absolute difference between the average age and a winner's age of less than 1 year, but many tournaments also had 10, 15, and even 25-year differences. Yet, when calculating mean and median estimates across tournaments, the variance seemed to average out. Throughout almost the entire tournament, the mean and median age differences seemed to remain at approximately six to seven years, indicating no trend towards greater homogeneity.

We repeated these analyses when investigating how the gender composition changed throughout the tournaments. In total, 17 males won the tournament (57%). Here again, demographics do not seem to play a major role in the tournaments. Most tournaments began with

50% males and 50% females. While there were certainly exceptions where the final handful of contestants were all one gender, the data in general showed that the final groups of contestants tended to include a mixture of both males and females. In summary, neither age nor gender of fellow contestants reflected in the demographics of the ultimate winner beyond the probabilities based on the characteristics of the entire sample (e.g. at the beginning of the tournament).

**Research questions 2 and 3.** Our second research question asked, can first and second place finishers be predicted by formed alliances? In other words, do voting blocs tend to eliminate rival voting blocs first and then proceed to turn on each other? Here, using the previously described Voting blocs (B = 0.178, S.E. = .06; $p$ = .003), we found that the size of one's voting bloc at the merge was positively correlated with one's final tournament rank. This suggests that voting blocs appear to strategically eliminate rivals giving all group members a better chance for success. Our third and final research question asked, does early group cohesion play a role in the success of tournament winners? Again, using the previously described early group cohesion. We found that early group cohesion was not associated with one's final rank (B = -0.381, S.E. = .58; $p$ = .511). We will discuss the results of the research questions in greater detail, including theoretical implications, in the conclusion.

## Discussion

Prior to the current work, tournament theory has been conceptualized as a competitive framework where contestants pursue top prizes with no regard for others. What we found is much more in line with Nash economics where cooperation—even when temporary—can be an alternative and supplementary path to success. Perhaps most notably, past research has treated tournament contestants as non-interacting entities and thus, oversimplifies the actual tournament dynamics [8]. Past scholars have argued that social network theory held "promise for reframing the questions we ask about tournaments" [5, p. 36]. The current research sought to examine the role of social networks in tournament-style promotion and reward systems that had not been empirically addressed. What was uncovered was a better understanding of how contestants advance through tournaments when social relationships play a role. In the current section, we highlight these contributions with an emphasis on how tournament theory is advanced by our work and how organizations can benefit from this research.

First, the current work combined the tournament theory literature and social networks literature, something that had been absent until this research. We found evidence to support the importance of the *contest-mobility* framework of job success, or that individuals advance of their own merit, while also examining the central role that social networks play in a tournament setting. In other words, this study indicates the importance of tie strength in tournament contests ($r$ = .12 between tie strength and tournament rank; $r$ = .44 between tie strength and rank when pre- and post-merge data are used) as well as performance ($r$ = .17 between performance and tournament rank) and supports the notion that people must get along to get ahead even when competing against each other in a tournament setting. This finding extends past knowledge of tournaments where contestants were non-interacting entities and tournament advancement was tied exclusively to individual performance. Yet, we did not find evidence of an interactive effects of having strong ties in addition to strong performance which points to an interesting quandary for contestants in a tournament concerning how to manage the competing demands of performance and coworker relationships, which will be discussed in greater detail in the future research section.

Second, we examined whether tie strength mediated the relation between group dissimilarity and tournament rank. In this study, we considered age, gender, and race. Quite interestingly, these surface-level demographic characteristics did not predict advancement in the

tournament (see Table 1) nor did a contestant's general differences with the group (see Table 2). We now suggest that surface-level characteristics may not play a role in tournaments in which strong ties are formed and contestants are able to develop long lasting relationships over the course of the entire tournament. We find this result interesting and wonder if, perhaps, deep-level diversity characteristics play a stronger role, such as personality and values, though we have no way to test this and suggest the topic for future research. Again, we find this result of note particularly from the perspective of organizations looking to build and benefit from a more diverse workforce. Specifically, our research suggests that getting diverse parties to work together may be achieved with a culture based on competition with fellow employees. Though we have no clear theoretical explanation for this empirical observation, we again hope that future research can explore the topic in greater detail.

Third, we did not find that initial tie formation influenced the strength of ties. Contestants that were able to develop strong ties early on, and maintain those ties, were not able to advance further into the tournament. Fourth, we asked several research questions that contribute to our knowledge of tournaments where there are interacting entities. Groups did not become more homogeneous over time to better mirror the demographic characteristics of first and second place winners. That is, the level of initial diversity in terms of race, gender, and age largely appeared to remain constant as the tournaments progressed. This observation conflicts with traditional views in organizations that 'birds of feather flock together' or that people who are more similar demographically would work to have a more homogenous group and we discuss this further in our future research section [58, 59].

Fifth, we learned that in-groups and out-groups were created and were dynamic in these tournaments in which the larger "in-group" tended to eliminate the contestants from the "out-group" members prior to dissolving their own "in-group." This observation lends credibility to notions of politically driven tournament success and perhaps explains why, in our model, social networks are important to tournament success. Finally, we learned that early group cohesion did not play a role in the success of tournament winners. In other words, groups that initially were quite cohesive in very early stages did not necessarily provide an advantage to tournament winners as allegiances tended to change from the beginning of the tournament to end of the tournament.

## Implications for theory, practice, and policy

For organizations considering how to use tournaments strategically, our research has a number of implications. First, in tournament systems where contestants have equal formal power, group dissimilarity does not appear to result in adverse outcomes for women, racial minorities, or older competitors above the age of 40. Advancement for women and minorities in organizations is both a societal concern and of organizational interest [60–62] as is the investigation of how diversity can lead to positive outcomes for organizations. Past research has suggested that more diversity leads to conflict and that conflict could lead to better or worse outcomes based on how it was managed [63]. Further, organizations have been very vocal about the need to have representation of traditionally underrepresented groups across all levels of an organization, but have lamented that a pipeline of talented workers from across demographic backgrounds was not readily available [64]. Our results suggest that to overcome some of the barriers that exist for workers based on age, gender, or race, organizations could implement a tournament-based system whereby people work together and compete for promotions in a transparent way.

Our observation concerning the importance, or not, of demographic characteristics deserves greater attention as it appears that notions of homophily were not influencing the

Survivor results. One explanation could be that because the amount of time contestants spent together was such that surface-level diversity differences were mitigated overtime perhaps in comparison to deeper-level diversity characteristics. How could this influence organizations? In *Survivor*, contestants are taken as a given by each other and it is not questioned how/why individuals are chosen for the show. This is in contrast to traditional organizations that have a known process for recruiting, evaluating, and selecting employees and research suggests that many methods of hiring are fraught with the potential for discrimination. Further, in organizations a single manager likely has undue influence over an individual's performance evaluations and promotional level, unlike *Survivor* where peers are responsible for tournament advancement. If more organizations were to establish rigorous controls in the structural characteristics (e.g., selection and promotion) of their Human Resource Management (HRM) systems, perhaps similar gender and racial equality results could be achieved.

The third implication is that tournament theory explains one way to mitigate the principal-agent problem. However, if an organization decides to implement a tournament system its members need to be cognizant of the implications of social networks on individual performance [5]. Tournaments are ideal when absolute performance is difficult to observe and/or evaluate. Consequently, relative performance is one means to determine promotion and allocate rewards. However, as illustrated in the current study, social networks play an important role in tournament style promotion and rewards systems. Thus, organizations must take care to consider the role of social networks in addition to performance when evaluating employees and making decisions about promotions to better ensure that rewards are allocated based on performance and that the best performing individuals are retained by organizations. Additionally, managers may encourage new employees to seek out mentors or join groups within the organization so as to nudge employees towards building their networks and maximizing the chances for future advancement.

Fourth, this study highlights the role of both "in-groups" and "out-groups" as well as the lesser role of initial cohesion. Initial cohesion did not appear to play a lasting role in the long-term success of tournament contestants. This may be of interest to organizations to know that initial starting locations of tournament contestants may not limit, or ultimately determine, advancement at least in terms of the relationships developed. Conversely, it is the formation of "in-groups" and "out-groups" that may be a little more concerning for organizations such that those in the "out-groups" have very little chance of advancing far in the tournament. In fact, it takes a very special performance for "out-group" individuals to advance to the end of the tournament. Conversely, more moderate performers could advance very far if one is in a larger, more dominate and cohesive group. Organizations should pay attention to the power of such "in-groups" and "out-groups" as it is unclear how these relate to individual motivation or the potential turnover of high performing individuals who are in the "out-group."

## Limitations and future directions

Our study examined a large dataset where individuals make strategic decisions about who to vote for and align themselves with to win. Though this dataset uniquely allowed us to examine the role that social networks play in an actual tournament setting, there are some limitations that we hope could be addressed with future research. First, the format of the tournament is such that consecutive rounds occur very quickly, and contestants are having to make decisions over relatively short periods of time regarding their success. This contrasts with an organization where political alliances and career success could unfold over the course of years.

Second, the nature of the *Survivor* tournament is such that artificial elements are introduced into the environment for the sake of the television viewer and may not be symptomatic of a

tournament in the business setting. For example, tournament contestants are aware that their behaviors are viewed later by friends and family and may be judged on social media. Third, for contestants on *Survivor*, there are no alternative tournaments or alternative roads to success. In an organization, individuals can leave and get a job at another organization, which is not the case for *Survivor*.

Fourth and finally, the isolation of the contestants of the show is clearly unique. In a real-world setting, individuals would be able to find background information about coworkers and use that information to make network relevant decisions (e.g., to work on a project with an individual). Thus, we think that using the *Survivor* dataset to study how social networks contribute to the success of individuals in tournaments is a unique, compelling, and interesting way to study the phenomena where we have far more transparency into contestant interactions than we would in a traditional business environment. However, we feel that our results need to be corroborated with future research that takes place in a tournament with higher fidelity to a business.

## Conclusion

Tournament theory is a popular conceptual framework well known across many academic and professional areas [5]. Yet past research into tournament theory has tended to evaluate rank-order success of tournament contestants while treating them as independent or non-interacting entities. Conversely, the current research examined the role of social networks in tournament-style and promotion research systems. We found that group dissimilarity and initial tie formation tended to play little if any role in influencing contestants' success in the tournaments. The most successful tournament contestants were not necessarily those that had strong performance, but were those that had strong performance and a strong social network in the form of strong ties. We concluded by discussing our findings as well as the theoretical and practical implications for tournament-style promotion systems.

## Author Contributions

**Conceptualization:** George C. Banks, Christopher E. Whelpley, Eean R. Crawford, Ernest H. O'Boyle, Sven Kepes.

**Data curation:** George C. Banks, Christopher E. Whelpley.

**Formal analysis:** George C. Banks, Eean R. Crawford.

**Methodology:** George C. Banks, Ernest H. O'Boyle.

**Supervision:** George C. Banks, Ernest H. O'Boyle.

**Validation:** George C. Banks.

**Writing – original draft:** George C. Banks, Christopher E. Whelpley.

**Writing – review & editing:** George C. Banks, Christopher E. Whelpley, Ernest H. O'Boyle, Sven Kepes.

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
