## [Decision Letter · Decision Letter 0]

12 Apr 2021

PONE-D-21-01453

Getting along to get ahead? The role of social context in tournament promotion and reward systems

PLOS ONE

Dear Dr. Banks,

Thank you for submitting your manuscript to PLOS ONE. After careful consideration, we feel that it has merit but does not fully meet PLOS ONE’s publication criteria as it currently stands. Therefore, we invite you to submit a revised version of the manuscript that addresses the points raised during the review process.

We look forward to receiving your revised manuscript.

Kind regards,

Zulqurnain Ali, PhD

Academic Editor

PLOS ONE

Journal Requirements:

5. Please amend either the title on the online submission form (via Edit Submission) or the title in the manuscript so that they are identical.

Additional Editor Comments:

Dear author(s),

I have now received reviewers’ comments on the your manuscript. Please find the reviewer comments at the bottom of this letter.

The reviewers show positive remarks about your manuscript, but they still suggest a few minor revisions. I recommend the authors to carefully address the comments and submit a revised version with a detail response of each of the comment. If you cannot address any particular comment, please explain properly in the response letter to reviewers. Good Luck!

Reviewers' comments:

Reviewer's Responses to Questions

**Comments to the Author**

1. Is the manuscript technically sound, and do the data support the conclusions?

Reviewer #1: Yes

Reviewer #2: Yes

Reviewer #3: Yes

2. Has the statistical analysis been performed appropriately and rigorously? 

Reviewer #1: Yes

Reviewer #2: Yes

Reviewer #3: Yes

3. Have the authors made all data underlying the findings in their manuscript fully available?

Reviewer #1: Yes

Reviewer #2: Yes

Reviewer #3: Yes

4. Is the manuscript presented in an intelligible fashion and written in standard English?

Reviewer #1: Yes

Reviewer #2: Yes

Reviewer #3: Yes

5. Review Comments to the Author

Reviewer #1: I found this study informative, which presents a new idea entitled, "Getting along to get ahead? The role of social context in tournament promotion and reward systems." This study explains that the role of social networks in tournament-style promotion and reward systems. Specifically, we seek to identify the importance of social relationships, such as group dissimilarity, initial tie formation, and tie strength in predicting tournament success. Bringing two largely independent research streams together (one focused on cooperation and one framed around competition), the study examines how individuals’ performance interacts with their social relationships—their social networks—to influence their chances of winning a tournament.

Abstract and Introduction improvement:

I am glad to assess this informative study. In my opinion, I have some guidelines for the authors to enhance the study quality before endorsing it for publication. As the Abstract is the main door or "FACE" of the manuscript, it should briefly present high-quality English with new information. I am recommending the authors of this study to expand Abstract, as it is too short. The Abstract should be around 250 words. I have suggested some studies to check the abstracts and improve yours and cite them in the introduction and build your study objectives like these studies.

Toqeer, Samia, Muhammad Aqeel, Kanwar Hamza Shuja, Akhtar Bibi, and Jaffar Abbas. 2021. "Attachment Styles, Facebook Addiction, Dissociation and Alexithymia in University Students; A Mediational Model." Nature-Nurture Journal of Psychology 1 (1):28-37. doi: http://thenaturenurture.org/index.php/psychology/article/view/2.

Abbas, J., Raza, S., Nurunnabi, M., Minai, M. S., & Bano, S. (2019). The Impact of Entrepreneurial Business Networks on Firms’ Performance Through a Mediating Role of Dynamic Capabilities. Sustainability, 11(11). https://doi.org/10.3390/su11113006

NeJhaddadgar, N., A. Ziapour, G. Zakkipour, J. Abbas, M. Abolfathi, and M. Shabani. 2020. "Effectiveness of telephone-based screening and triage during COVID-19 outbreak in the promoted primary healthcare system: a case study in Ardabil province, Iran." Z Gesundh Wiss:1-6. doi: 10.1007/s10389-020-01407-8.

Literature section

It presents a good summary of the literature. I suggest authors add the literature as recommended below to improve the manuscript. Overall, the authors have creatively linked variables. It reflects an innovative model of the study. I am pleased to read this article. However, I have some suggestions for the authors to enhance the quality of the literature section. The authors can add few lines about technological innovations and environmental responsibility practices. Please see the suggested studies and cite them to enhance the literature section.

Lebni, J. Y., R. Toghroli, J. Abbas, N. NeJhaddadgar, M. R. Salahshoor, M. Mansourian, H. D. Gilan, N. Kianipour, F. Chaboksavar, S. A. Azizi, and A. Ziapour. 2020. "A study of internet addiction and its effects on mental health: A study based on Iranian University Students." J Educ Health Promot 9:205. doi: 10.4103/jehp.jehp_148_20.

Abbas, J., Aman, J., Nurunnabi, M., & Bano, S. (2019). The Impact of Social Media on Learning Behavior for Sustainable Education: Evidence of Students from Selected Universities in Pakistan. Sustainability, 11(6). https://doi.org/10.3390/su11061683

Abbas, J., Zhang, Q., Hussain, I., Akram, S., Afaq, A., & Shad, M. A. (2020). Sustainable Innovation in Small Medium Enterprises: The Impact of Knowledge Management on Organizational Innovation through a Mediation Analysis by Using SEM Approach. Sustainability, 12(6). https://doi.org/10.3390/su12062407

Methods and Results

The results section of the paper presents a good view of the study. This work presents a notable investigation on a selected topic. I suggest including some graphical presentations to improve the quality of this study. Please see the proposed studies and see the graphical representation. Improve your work like these studies and cite them in this section.

Fattahi, E., M. Solhi, J. Abbas, P. Kasmaei, S. Rastaghi, M. Pouresmaeil, A. Ziapour, and H. D. Gilan. 2020. "Prioritization of needs among students of University of Medical Sciences: A needs assessment." J Educ Health Promot 9 (1):57. doi: 10.4103/0445-7706.281641.

Moradi, F., S. Tourani, A. Ziapour, J. Abbas, M. Hematti, E. J. Moghadam, A. Aghili, and A. Soroush. 2020. "Emotional Intelligence and Quality of Life in Elderly Diabetic Patients." Int Q Community Health Educ:272684X20965811. doi: 10.1177/0272684X20965811.

Yoosefi Lebni, J., J. Abbas, F. Khorami, B. Khosravi, A. Jalali, and A. Ziapour. 2020. "Challenges Facing Women Survivors of Self-Immolation in the Kurdish Regions of Iran: A Qualitative Study." Front Psychiatry 11 (778):778. doi: 10.3389/fpsyt.2020.00778.

Conclusion

I suggest you make a separate heading of the conclusion and do not mix it with implications.

Policy Recommendations

I again recommend you to make a separate heading of the Policy Recommendations.

The conclusion section is acceptable. Overall, this presents a good piece of research work. I recommend that authors do a little more work and revise this article accordingly. I suggest the authors check English quality and fix some weak sentences. If you have already taken English editing service, ask them to recheck the quality to meet scientific merit for publication. I endorse this manuscript for publication after minor corrections, as suggested.

Reviewer #2: I am glad to review and assess this interesting article .Overall, the manuscript is a good piece of work. The English level is good and smooth, e.g., the language standard, specifically the grammar, of sufficient quality to meet scientific merit for publication. I accept this manuscript as I have recommended.

Reviewer #3: This article is interesting and its quality is good. I have few concerns

1. The article is long, I notice few repetitions of the same ideas. Authors should try to be concise and avoid unnecessary repetitions without affecting the quality of the contents.

2. The references are quite old. The current state of the research is not apparent in the article. Authors should cite latest articles in the field

3. The authors mix two citation formats. Please follow the journal citation style.

6. PLOS authors have the option to publish the peer review history of their article (what does this mean?). If published, this will include your full peer review and any attached files.

Reviewer #1: No

Reviewer #2: No

Reviewer #3: No

---

## [Author Response · Author response to Decision Letter 0]

14 May 2021

Additional Editor Comments:

Dear author(s),

I have now received reviewers’ comments on the your manuscript. Please find the reviewer comments at the bottom of this letter.

The reviewers show positive remarks about your manuscript, but they still suggest a few minor revisions. I recommend the authors to carefully address the comments and submit a revised version with a detail response of each of the comment. If you cannot address any particular comment, please explain properly in the response letter to reviewers. Good Luck!

Author response

Thank you for the opportunity to revise and resubmit our manuscript. We have made revisions to our manuscript and responded point by point to the reviewers. We appreciated their positive support. There were a number of references proposed by Reviewer 1 that we were unable to edit as they pertained to entirely differently content and methodological areas. That being said, we were able to make changes in response to the rest of the comments.

Reviewer 1, Comment 1

Reviewer #1: I found this study informative, which presents a new idea entitled, "Getting along to get ahead? The role of social context in tournament promotion and reward systems." This study explains that the role of social networks in tournament-style promotion and reward systems. Specifically, we seek to identify the importance of social relationships, such as group dissimilarity, initial tie formation, and tie strength in predicting tournament success. Bringing two largely independent research streams together (one focused on cooperation and one framed around competition), the study examines how individuals’ performance interacts with their social relationships—their social networks—to influence their chances of winning a tournament.

Author response

We are happy to hear that you like the title and abstract.

Reviewer 1, Comment 2

Abstract and Introduction improvement:

I am glad to assess this informative study. In my opinion, I have some guidelines for the authors to enhance the study quality before endorsing it for publication. As the Abstract is the main door or "FACE" of the manuscript, it should briefly present high-quality English with new information. I am recommending the authors of this study to expand Abstract, as it is too short. The Abstract should be around 250 words. I have suggested some studies to check the abstracts and improve yours and cite them in the introduction and build your study objectives like these studies.

Toqeer, Samia, Muhammad Aqeel, Kanwar Hamza Shuja, Akhtar Bibi, and Jaffar Abbas. 2021. "Attachment Styles, Facebook Addiction, Dissociation and Alexithymia in University Students; A Mediational Model." Nature-Nurture Journal of Psychology 1 (1):28-37. doi: http://thenaturenurture.org/index.php/psychology/article/view/2.

Abbas, J., Raza, S., Nurunnabi, M., Minai, M. S., & Bano, S. (2019). The Impact of Entrepreneurial Business Networks on Firms’ Performance Through a Mediating Role of Dynamic Capabilities. Sustainability, 11(11). https://doi.org/10.3390/su11113006

NeJhaddadgar, N., A. Ziapour, G. Zakkipour, J. Abbas, M. Abolfathi, and M. Shabani. 2020. "Effectiveness of telephone-based screening and triage during COVID-19 outbreak in the promoted primary healthcare system: a case study in Ardabil province, Iran." Z Gesundh Wiss:1-6. doi: 10.1007/s10389-020-01407-8.

Author response

Thank you for this comment. We have revised our Abstract and expanded the length. Regarding the recommended citations above and below, members of the authorship team independently read these articles and found them very interesting and well-conducted, but somewhat tangential to our study’s focus on social networks and tournaments. Perhaps we missed the thread tying these works to ours. Thus, if given the opportunity to further revise our work we would appreciate if you would elaborate on where you see these articles supporting our theory and findings. 

Reviewer 1, Comment 3

Literature section

It presents a good summary of the literature. I suggest authors add the literature as recommended below to improve the manuscript. Overall, the authors have creatively linked variables. It reflects an innovative model of the study. I am pleased to read this article. However, I have some suggestions for the authors to enhance the quality of the literature section. The authors can add few lines about technological innovations and environmental responsibility practices. Please see the suggested studies and cite them to enhance the literature section.

Lebni, J. Y., R. Toghroli, J. Abbas, N. NeJhaddadgar, M. R. Salahshoor, M. Mansourian, H. D. Gilan, N. Kianipour, F. Chaboksavar, S. A. Azizi, and A. Ziapour. 2020. "A study of internet addiction and its effects on mental health: A study based on Iranian University Students." J Educ Health Promot 9:205. doi: 10.4103/jehp.jehp_148_20.

Abbas, J., Aman, J., Nurunnabi, M., & Bano, S. (2019). The Impact of Social Media on Learning Behavior for Sustainable Education: Evidence of Students from Selected Universities in Pakistan. Sustainability, 11(6). https://doi.org/10.3390/su11061683

Abbas, J., Zhang, Q., Hussain, I., Akram, S., Afaq, A., & Shad, M. A. (2020). Sustainable Innovation in Small Medium Enterprises: The Impact of Knowledge Management on Organizational Innovation through a Mediation Analysis by Using SEM Approach. Sustainability, 12(6). https://doi.org/10.3390/su12062407

Author response

We thank you for the feedback regarding the literature section of the manuscript. Both you and Reviewer 3 suggested a number of improvements and we have revised our introduction to include more supporting and recent citations. As for the technological innovations and environmental responsibility practices citations, we again found them to be interesting and well-grounded. However, we also found ourselves not quite sure how to integrate the topics into our manuscript in a clear way. Your guidance would be greatly appreciated into how to leverage this supporting material. 

Reviewer 1, Comment 4

Methods and Results

The results section of the paper presents a good view of the study. This work presents a notable investigation on a selected topic. I suggest including some graphical presentations to improve the quality of this study. Please see the proposed studies and see the graphical representation. Improve your work like these studies and cite them in this section.

Fattahi, E., M. Solhi, J. Abbas, P. Kasmaei, S. Rastaghi, M. Pouresmaeil, A. Ziapour, and H. D. Gilan. 2020. "Prioritization of needs among students of University of Medical Sciences: A needs assessment." J Educ Health Promot 9 (1):57. doi: 10.4103/0445-7706.281641.

Moradi, F., S. Tourani, A. Ziapour, J. Abbas, M. Hematti, E. J. Moghadam, A. Aghili, and A. Soroush. 2020. "Emotional Intelligence and Quality of Life in Elderly Diabetic Patients." Int Q Community Health Educ:272684X20965811. doi: 10.1177/0272684X20965811.

Yoosefi Lebni, J., J. Abbas, F. Khorami, B. Khosravi, A. Jalali, and A. Ziapour. 2020. "Challenges Facing Women Survivors of Self-Immolation in the Kurdish Regions of Iran: A Qualitative Study." Front Psychiatry 11 (778):778. doi: 10.3389/fpsyt.2020.00778.

Author response

Thanks for your comment. We are happy to incorporate additional graphical representations, but note that we already include a representation of the social networks, the interaction plot, as well as the theoretical model. We are also mindful that reviewer 3 asked us to streamline the manuscript in terms of length. If there are any specific graphical representations that you believe would enhance or clarify our study, we would be happy to incorporate them. 

Reviewer 1, Comment 5

Conclusion

I suggest you make a separate heading of the conclusion and do not mix it with implications.

Author response

We now have a standalone Conclusion section that is on page 35.

Reviewer 1, Comment 6

Policy Recommendations

I again recommend you to make a separate heading of the Policy Recommendations.

The conclusion section is acceptable. Overall, this presents a good piece of research work. I recommend that authors do a little more work and revise this article accordingly. I suggest the authors check English quality and fix some weak sentences. If you have already taken English editing service, ask them to recheck the quality to meet scientific merit for publication. I endorse this manuscript for publication after minor corrections, as suggested.

Author response

Thank you very much for this insight. Based on your guidance, we have repositioned the previous Implications section as Implications for Theory, Practice, and Policy. We have performed a careful check of the English writing. Thank you for your endorsement!

Reviewer #2, Comment 1

I am glad to review and assess this interesting article .Overall, the manuscript is a good piece of work. The English level is good and smooth, e.g., the language standard, specifically the grammar, of sufficient quality to meet scientific merit for publication. I accept this manuscript as I have recommended.

Author response

Thank you for your positive response.

Reviewer #3, Comment 1

: This article is interesting and its quality is good. I have few concerns

1. The article is long, I notice few repetitions of the same ideas. Authors should try to be concise and avoid unnecessary repetitions without affecting the quality of the contents.

Author response

Thank you for raising this suggestion. We have streamlined the paper by cutting 2 pages. We would be happy to further distill the message if you see specific opportunities for additional improvement by subtraction. 

Reviewer #3, Comment 2

2. The references are quite old. The current state of the research is not apparent in the article. Authors should cite latest articles in the field

Author response

Thank you very much. This is a timely topic area that generates new research almost daily. We reviewed a variety of research, commentaries, and literature reviews published on the topic since 2019. After a thorough vetting process of the key citations that are most relevant and most inform our research question and results, we added the following references;

Arend, R. J. 2019. Cheat to win: How to hack tournament theory. Business Research Quarterly, 22(4): 216-225.

Ekinci, E., Kauhanen, A., & Waldman, M. 2019. Bonuses and promotion tournaments: Theory and evidence. The Economic Journal, 129(622): 2342-2389.

Guiteierrez, C., Obloj, T., & Frank, D. H. (2020). Better to have led and lost than never to have led at all? Lost leadership and effort provision in dynamic tournaments. Strategic Management Journal, 42, 774-801.

Li, J., Shi, W., Connelly, B., Yi, X., & Qin, X. (in press). CEO Awards and Financial Misconduct. Journal of Management.

Pruijssers, J. L., Heugens, P. P., & Van Oosterhout, J. (2020). Winning at a losing game? Side-effects of perceived tournament promotion incentives in audit firms. Journal of Business Ethics, 162(1), 149-167.

Reviewer #3, Comment 3

3. The authors mix two citation formats. Please follow the journal citation style.

Author response

Thank you for pointing this out and we apologize for the inconsistency. We have performed a careful check of the citation and reference format. The format is now fully aligned with the PLoS One requirements.

---

## [Editor Report · Decision Letter 1]

25 May 2021

PONE-D-21-01453R1

Getting along to get ahead: The role of social context in tournament promotion and reward systems

PLOS ONE

Dear Dr. Banks,

Thank you for submitting your manuscript to PLOS ONE. After careful consideration, we feel that it has merit but does not fully meet PLOS ONE’s publication criteria as it currently stands. Therefore, we invite you to submit a revised version of the manuscript that addresses the points raised during the review process.

We look forward to receiving your revised manuscript.

Kind regards,

Zulqurnain Ali, PhD

Academic Editor

PLOS ONE

Journal Requirements:

Additional Editor Comments (if provided):

Please read respond to the following reviewers comments:

his article is interesting and its quality is good. I have few concerns

1. The article is long, I notice few repetitions of the same ideas. Authors should try to be concise and avoid unnecessary repetitions without affecting the quality of the contents.

2. The references are quite old. The current state of the research is not apparent in the article. Authors should cite latest articles in the field

3. The authors mix two citation formats. Please follow the journal citation style.

4. Proofread your paper from an English copy editor to improve the quality of language across the manuscript.

---

## [Author Response · Author response to Decision Letter 1]

23 Jun 2021

Editor Opening comment

Dear Dr. Banks,

Thank you for submitting your manuscript to PLOS ONE. After careful consideration, we feel that it has merit but does not fully meet PLOS ONE’s publication criteria as it currently stands. Therefore, we invite you to submit a revised version of the manuscript that addresses the points raised during the review process.

Kind regards,

Zulqurnain Ali, PhD

Academic Editor

PLOS ONE

Author response

Thank you again for the opportunity to revise our submitted work.

We look forward to receiving your revised manuscript.

Additional Editor Comments (if provided):

Please read respond to the following reviewers comments:

This article is interesting and its quality is good. I have few concerns

Reviewer comment

1. The article is long, I notice few repetitions of the same ideas. Authors should try to be concise and avoid unnecessary repetitions without affecting the quality of the contents.

Author response

We streamlined the original submission by 2 pages with an emphasis on removing any redundancy. If there is a particular area where you are concerned about redundancy please let us know.

Reviewer comment

2. The references are quite old. The current state of the research is not apparent in the article. Authors should cite latest articles in the field

Author response

We have added 5 references to current work since our original submission that includes articles which are from 2019, 2020, or in press.

Arend, R. J. 2019. Cheat to win: How to hack tournament theory. Business Research Quarterly, 22(4): 216-225.

Ekinci, E., Kauhanen, A., & Waldman, M. 2019. Bonuses and promotion tournaments: Theory and evidence. The Economic Journal, 129(622): 2342-2389.

Guiteierrez, C., Obloj, T., & Frank, D. H. (2020). Better to have led and lost than never to have led at all? Lost leadership and effort provision in dynamic tournaments. Strategic Management Journal, 42, 774-801.

Li, J., Shi, W., Connelly, B., Yi, X., & Qin, X. (in press). CEO Awards and Financial Misconduct. Journal of Management.

Pruijssers, J. L., Heugens, P. P., & Van Oosterhout, J. (2020). Winning at a losing game? Side-effects of perceived tournament promotion incentives in audit firms. Journal of Business Ethics, 162(1), 149-167.

Reviewer comment

3. The authors mix two citation formats. Please follow the journal citation style.

Author response

We have checked our citation format against the style guide for PLOS One to ensure consistency in the journal citation style.

Reviewer comment

4. Proofread your paper from an English copy editor to improve the quality of language across the manuscript.

Author response

We have performed an additional review of our manuscript for grammar and quality of language.

---

## [Editor Report · Decision Letter 2]

1 Sep 2021

Getting along to get ahead: The role of social context in tournament promotion and reward systems

PONE-D-21-01453R2

Dear Dr. Banks,

We’re pleased to inform you that your manuscript has been judged scientifically suitable for publication and will be formally accepted for publication once it meets all outstanding technical requirements.

Kind regards,

Zulqurnain Ali, PhD

Academic Editor

PLOS ONE
---

## [Editor Report · Acceptance letter]

7 Sep 2021

PONE-D-21-01453R2 

Getting along to get ahead: The role of social context in tournament promotion and reward systems 

Dear Dr. Banks:

I'm pleased to inform you that your manuscript has been deemed suitable for publication in PLOS ONE. Congratulations! Your manuscript is now with our production department. 

Kind regards, 

on behalf of

Dr. Zulqurnain Ali 

Academic Editor

PLOS ONE